# SYNTACTIC REPRESENTATIONS IN THE HUMAN BRAIN: BEYOND EFFORT-BASED METRICS

## ABSTRACT

We are far from having a complete mechanistic understanding of the brain computations involved in language processing and of the role that syntax plays in those computations. Most language studies do not computationally model syntactic structure and most studies that do model syntactic processing use effort-based metrics. These metrics capture the effort needed to process the syntactic information given by every word (Brennan et al., 2012; Hale et al., 2018; Brennan et al., 2016). They can reveal *where* in the brain syntactic processing occurs, but not *what* features of syntax are processed by different brain regions. Here, we move beyond effort-based metrics and propose explicit features capturing the syntactic structure that is incrementally built while a sentence is being read. Using these features and functional Magnetic Resonance Imaging (fMRI) recordings of participants reading a natural text, we study the brain representation of syntax. We find that our syntactic structure-based features are better than effort-based metrics at predicting brain activity in various parts of the language system. We show evidence of the brain representation of complex syntactic information such as phrase and clause structures. We see that regions well-predicted by syntactic features are distributed in the language system and are not distinguishable from those processing semantics. Our results call for a shift in the approach used for studying syntactic processing.

## 1 INTRODUCTION

Neuroscientists have long been interested in how the brain processes syntax. To date, there is no consensus on which brain regions are involved in processing it. Classically, only a small number of regions in the left hemisphere were thought to be involved in language processing. More recently, the language system was proposed to involve a set of brain regions spanning the left and right hemisphere (Fedorenko & Thompson-Schill, 2014). Similarly, some findings show that syntax is constrained to specific brain regions (Grodzinsky & Friederici, 2006; Friederici, 2011), while other findings show syntax is distributed throughout the language system (Blank et al., 2016; Fedorenko et al., 2012; 2020).

The biological basis of syntax was first explored through studies of the impact of brain lesions on language comprehension or production (Grodzinsky, 2000) and later through non-invasive neuroimaging experiments that record brain activity while subjects perform language tasks, using methods such as functional Magnetic Resonance Imaging (fMRI) or electroencephalography (EEG). These experiments usually isolate syntactic processing by contrasting the activity between a difficult syntactic condition and an easier one and by identifying regions that increase in activity with syntactic effort (Friederici, 2011). An example of these conditions is reading a sentence with an object-relative clause (e.g. "The rat *that the cat chased* was tired"), which is more taxing than reading a sentence with a subject-relative clause (e.g. "The cat *that chased the rat* was tired"). In the past decade, this approach was extended to study syntactic processing in naturalistic settings such as when reading or listening to a story (Brennan et al., 2012; Hale et al., 2018; Willems et al., 2015). Because such complex material is not organized into conditions, neuroscientists have instead devised effort-based metrics capturing the word-by-word evolving syntactic demands required to understand the material. Brain regions with activity correlated with those metrics are suggested to be involved in processing syntax.

We use the term effort-based metrics to refer to uni-dimensional measures capturing word-by-word syntactic demands. A standard approach for constructing a syntactic effort-based metric is to assume

a sentence's syntactic representation and estimate the number of syntactic operations performed at each word. Node Count is popular such metric. It relies on constituency trees (structures that capture the hierarchical grammatical relationship between the words in a sentence). While traversing the words of the sentence in order, subtrees of the constituency tree get completed; Node Count refers to the number of such subtrees that get completed at each word, effectively capturing syntactic load or effort. Brennan et al. (2012) use Node Count to support the theory that the Anterior Temporal Lobe (ATL) is involved in syntactic processing. Another example of an effort-based metric is given by an EEG study by Hale et al. (2018). They show that parser action count (the number of possible actions a parser can take at each word) is predictive of the P600, a positive peak in the brain's electrical activity occurring around 600ms after word onset. The P600 is hypothesized to be driven by syntactic processing (to resolve incongruencies), and the results of Hale et al. (2018) align with this hypothesis.

Though effort-based metrics are a good proposal for capturing the effort involved in integrating a word into the syntactic structure of a sentence, they are not reflective of the entire syntactic information in play. Hence, these metrics cannot be used to study the brain representation of syntactic constructs such as nouns, verbs, relationships and dependencies between words, and the complex hierarchical structure of phrases and sentences.

Constituency trees and dependency trees are the two main structures that capture a sentence's syntactic structure. Constituency trees are derived using phrase structure grammars that encode valid phrase and clause structure (see Figure 1(A) for an example). Dependency trees encode relations between pairs of words such as subject-verb relationships. We use representations derived from both types of trees. We derive word level dependency (DEP) labels from dependency trees, and we focus on encoding the structural information given by constituency trees since we want to analyze if the brain builds hierarchical representations of phrase structure. We characterize the syntactic structure inherent in sentence constituency trees by computing an evolving vector representation of the syntactic structure processed at each word using the subgraph embedding algorithm by Adhikari et al. (2018). We show that our syntactic structure embeddings – along with other simpler syntactic structure embeddings built using conventional syntactic features such as part-of-speech (POS) tags and DEP tags – are better than effort-based metrics at predicting the fMRI data of subjects reading text. This indicates that representations of syntax, and not just syntactic effort, can be observed in fMRI.

We also address the important question of whether regions that are predicted by syntactic features are selective for syntax, meaning they are only responsive to syntax and not to other language properties such as semantics. To answer this question, we model the semantic properties of words using a contextual word embedding space (Devlin et al., 2018). We find that regions that are predicted by syntactic features are also predicted by semantic features and thus are not selective for syntax.

**Scientific questions** We ask three main questions:
- How can scientists construct syntactic structure embeddings that capture the syntactic structure inherent in phrases and sentences?
- Are these embeddings better at predicting brain activity compared to effort-based metrics when used as inputs to encoding models?
- Which brain regions are involved in processing complex syntactic structure and are they different from regions involved in semantic processing?

**Contributions** We make four main contributions:
- We propose a subgraph embeddings-based method to model the syntactic structure inherent in phrases and sentences.
- We show that effort-based metrics can be complemented by syntactic structure embeddings which can predict brain activity to a larger extent than effort-based metrics.
- Using our syntactic structure embeddings, we find some evidence supporting the hypothesis that the brain processes and represents complex syntactic information such as phrase and clause structure.
- We find evidence supporting the existing hypothesis that syntactic processing appears to be distributed in the language network in regions that are not selective for syntax.

## 2 METHODS

We first describe the syntactic features used in this study and their generation. All of the features we use are incremental i.e. they are computed per word. We then describe our fMRI data analyses.

**Effort-based metrics** We use four effort-based metrics in our analyses - Node Count, Syntactic Surprisal, word frequency and word length. Node Count is an effort-based metric popular in neuroscience. To compute it, we obtain the constituency tree of each sentence using the self-attentive encoder-based constituency parser by Kitaev & Klein (2018). We compute Node Count for each word as the number of subtrees that are completed by incorporating this word into its sentence. Syntactic Surprisal is another effort-based metric proposed by Roark et al. (2009) and is computed using an incremental top down parser (Roark, 2001). Both of these metrics aim to measure the amount of effort that is required to integrate a word into the syntactic structure of its sentence. The word frequency metric is computed using the wordfreq package (Speer et al., 2018) as the Zipf frequency of a word. This is the base-10 logarithm of the number of occurrences per billion of a given word in a large text corpus. Finally, word length is the number of characters in the presented word. The last two metrics approximate the amount of effort that is required to read a word.

**Constituency tree-based Graph Embeddings (ConTreGE)** Constituency trees are a rich source of syntactic information. We build three representations of these trees that encode this information:

**(a)** The largest subtree which is completed upon incorporating a word into a sentence (see figure 1(B)) is representative of the implicit syntactic information given by the word. Given that Node Count reduces all of the information present in these subtrees to just one number, it is easy to see that it cannot effectively capture this information. POS tags (categorize words into nouns, verbs, adjectives, etc.) also capture some of the information present in these trees as they encode phrase structure to a certain extent. But, they are incapable of completely encoding their hierarchical structure and the parsing decisions which are made while generating them. In order to better encode their structure, we first build subgraph embeddings of these completed subtrees called ConTreGE Comp vectors.

**(b)** We hypothesize that the brain not only processes structure seen thus far but also predicts future structure from structure it already knows. To test this, we construct embeddings, simply called ConTreGE vectors, using incomplete subtrees that are constructed by retaining all the phrase structure grammar productions that are required to derive the words seen till now, thereby allowing us to capture future sentence structure (in the form of future constituents) before the full sentence is read (see figure 1 (C)). These subtrees contain leaves that are non-terminal symbols unlike complete subtrees that only have terminal symbols (words and punctuation) as leaves. In this context, a non-terminal symbol is a symbol that can be derived further using some rule in the phrase structure grammar (ex. NP, VP, etc.). If incomplete subtrees are more representative of the brain's processes, it would mean that the brain expects certain phrase structures even before the entire phrase or sentence is read. ConTreGE Comp and ConTreGE vectors need to be built using accurate constituency trees constructed using the whole sentence. Thus, we reuse the trees generated to compute Node Count to build them.

**(c)** Further, the brain could be computing several possible top down partial parses that can derive the words seen thus far (see figures 1 (D) and (E)) and modifying the list of possible parses as future words are read. To test this hypothesis, we designed Incremental ConTreGE (InConTreGE) vectors that are representative of the most probable parses so far. For a given word, its InConTreGE vector is computed as: $v = \sum_{i=1}^{5} e^{-s_i} W_i$ where $W_i$ is the subgraph embedding of a partial parse tree built by an incremental top-down parser (Roark 2001 CoLing) after reading the word and $s_i$ is the score assigned to this partial parse that is inversely proportional to the parser's confidence in this tree.

To effectively capture the structure of all subtrees, we encode them using the subgraph embeddings proposed by Adhikari et al. (2018) which preserve the neighbourhood properties of subgraphs. A long fixed length random walk on a subgraph is generated to compute its embedding. Since consecutive nodes in a random walk are neighbours, a long walk can effectively inform us about the neighbourhoods of nodes in the subgraph. Each node in a walk is identified using its unique ID. So, a random walk can be interpreted as a "paragraph" where the words are the node IDs. Finally, the subgraph's embedding is computed as the Paragraph Vector (Le & Mikolov, 2014) of this paragraph that is representative of the subgraph's structure. Note that all of the subtrees of a given type (complete, incomplete or partial parse) are encoded together. This ensures that all ConTreGE Comp vectors, all ConTreGE vectors and all InConTreGE vectors are in our own spaces.

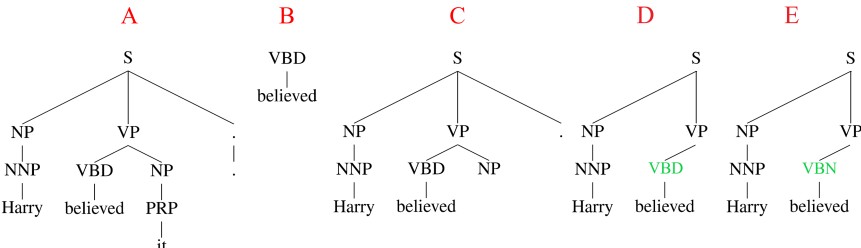

Figure 1: Example of complete and incomplete subtrees and two possible partial parses: Part A shows a sentence's constituency tree generated by a self-attentive encoder-based constituency parser (Kitaev & Klein, 2018) using all of its words. The largest completed subtree for "believed" is shown in part B and the incomplete subtree generated till "believed" is shown in part C. Incomplete subtrees are generally much deeper than complete ones. In parts D and E, we can see two possible partial parses generated by an incremental top-down parser (Roark, 2001) only using the words till "believed". We see that the POS tag assigned to "believed" is different in the two parses.

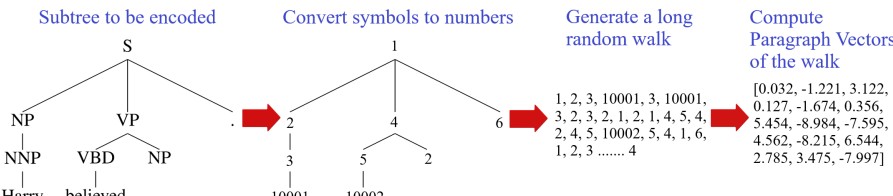

Figure 2: Steps for encoding subtrees.

Figure 2 illustrates the subtree encoding process. First, every unique non-terminal in the subtrees is mapped to a unique number (ex. S is mapped to 1, NP is mapped to 2, etc.) and every terminal is mapped to a unique number that is representative of the order in which they were presented (the first presented token is mapped to 10000, the second token is mapped to 10001 and so on). We did not map each unique terminal to a unique number (for instance, we did not map all instances of "Harry" to one number) because a random walk through the tree could give us word co-occurrence information and thus lead to the inclusion of some semantic information in the vectors.

Every tree node's label is then replaced by the number it was mapped to in the previous step. The edge lists of these subtrees are supplied to the subgraph embedding generation algorithm to finally obtain 15-dimensional vectors for every presented word. The length of the random walks is set to 100000 and we use an extension of the Distributed Bag of Nodes (DBON) model proposed by Le & Mikolov (2014) for generating Paragraph Vectors called Sub2Vec-DBON by Adhikari et al. (2018). The length of the sliding window is set to 5 and the model is trained for 20 epochs. Since ConTreGE Comp, ConTreGE and InConTreGE encode information about the neighbourhoods of all nodes in the constituency trees, they can capture their hierarchical structure. Thus, brain regions predicted by these vectors are likely to be involved in building and encoding hierarchical sentence structure.

**Punctuation** We create one-hot binary vectors indicating the type of punctuation that was presented along with a word (e.g. **.** or **,**). For example, a sentence might have ended with "Malfoy.". In this punctuation-based feature space, the column corresponding to **.** will be set to 1 for this word. While punctuation is seldom considered a syntactic feature, sentence boundaries are highly correlated with changes in working memory load. These changes are bound to be a great source of variability in the fMRI signal (as we will observe later). Failing to account for sentence boundaries and working memory might be a source of confounding that has been ignored in the literature.

**Part-of-speech tags and dependency tags** We use two standard word-level syntactic features - POS and DEP tags. The POS tag of a word is read off previously generated constituency trees. The DEP tag of a word (ex. subject, object, etc.) correspond to its assigned role in the dependency trees of the presented sentences which were generated using the spaCy English dependency parser (2). We create one-hot binary vectors indicating the POS tag and the DEP tag of each word and concatenate them to create one feature space which we refer to as simple syntactic structure embeddings.

**Semantic features** We adapt the vectors obtained from layer 12 of a pretrained (1) cased BERT-large model (Devlin et al., 2018) to identify regions that process semantics. We use layer 12 because of previous work showing that middle layers of sentence encoders are optimal for predicting brain activity (Jain & Huth, 2018; Toneva & Wehbe, 2019). We obtain the contextual embeddings for a word by running the pretrained model only on the words seen thus far, preventing the inclusion of future semantic information. Since a presented word can be broken up into multiple subtokens, we compute its embedding as the average of the subtokens' embeddings. Using principal component analysis (PCA), we reduce their dimensionality to 15 to match the ConTreGE vectors' dimensionality.

**fMRI data** We use the fMRI data of 9 subjects reading chapter 9 of *Harry Potter and the Sorcerer's Stone* (Rowling, 2012), collected and made available by Wehbe et al. (2014). Words are presented one at a time at a rate of 0.5s each. All the brain plots shown here are averages over the 9 subjects in the Montreal Neurological Institute (MNI) space. Preprocessing details are in Appendix B.

**Predicting brain activity** The applicability of a given syntactic feature in studying syntactic processing is determined by its efficacy in predicting the brain data described above. Ridge regression is used to perform these predictions and their coefficient of determination ($R^2$ score) measures the feature's efficacy. For each voxel of each subject, the regularization parameter is chosen independently. We use Ridge regression because of its computational efficiency and because of the Wehbe et al. (2015) results showing that with such fMRI data, as long as the regularization parameter is chosen by cross-validation for each voxel independently, different regularization techniques lead to similar results. Indeed, Ridge regression is a common regularization technique used for predictive fMRI models (Mitchell et al., 2008; Nishimoto et al., 2011; Wehbe et al., 2014; Huth et al., 2016).

For every voxel, a model is fit to predict the signals $Y = [y_1, y_2, \ldots, y_n]$ recorded in that voxel where $n$ is the number of time points (TR, or time to repetition). The words are first grouped by the TR in which they were presented. Then, the features of words in every group are summed to form a sequence of features $X = [x_1, x_2, \ldots, x_n]$ aligned with the brain signals. The response measured by fMRI is an indirect consequence of brain activity that peaks about 6 seconds after stimulus onset. A common solution to account for this delay is to express brain activity as a function of the features of the preceding time points (Nishimoto et al., 2011; Wehbe et al., 2014; Huth et al., 2016). Thus, we train our models to predict any $y_i$ using $x_{i-1}, x_{i-2}, x_{i-3}$ and $x_{i-4}$.

We test the models in a cross-validation loop: the data is first split into 4 contiguous and equal sized folds. Each model uses three folds of the data for training and one fold for evaluation. We remove the data from the 5 TRs which either precede or follow the test fold from the training set of folds. This is done to avoid any unintentional data leaks since consecutive $y_i$s are correlated with each other because of the lag and continuous nature of the fMRI signal. The brain signals and the word features which comprise the training and testing data for each model are individually Z-scored. After training we obtain the predictions for the validation fold. The predictions for all folds are concatenated (to form a prediction for the entire experiment in which each time point is predicted from a model trained without the data for that time point). Note that since all 3 ConTreGe vectors are stochastic, we construct them 5 times each, and learn a different model each time. The predictions of the 5 models are averaged together into a single prediction. The $R^2$ score is computed for every voxel using the predictions and the real signals.

We run a permutation test to test if $R^2$ scores are significantly higher than chance. We permute blocks of contiguous fMRI TRs, instead of individual TRs, to account for the slowness of the underlying hemodynamic response. We choose a common value of 10 TRs (Deniz et al., 2019). The predictions are permuted within fold 5000 times, and the resulting $R^2$ scores are used as an empirical distribution of chance performance, from which the p-value of the unpermuted performance is estimated. We also run a bootstrap test to test if a model has a higher $R^2$ score than another. The difference is that in each iteration, we permute (using the same indices) the predictions of both models and compute the difference of their $R^2$ and use the resulting distribution to estimate the p-value of the unpermuted difference. Finally, the Benjamni-Hochberg False Discovery Rate correction (Benjamini & Hochberg, 1995) is used for all tests (appropriate because fMRI data is considered to have positive dependence (Genovese, 2000)). The correction is performed by grouping together all the voxel-level $p$-values (i.e. across all subjects and feature groups) and choosing one threshold for all of our results. The correction is done in this way since we test multiple prediction models across multiple voxels and subjects. To compute Region of Interest (ROI) statistics, left-hemisphere ROI masks for the language

system obtained from a "sentence vs. non-word" fMRI contrast (Fedorenko et al., 2010) are obtained from (3) and mirrored to obtain the right-hemisphere ROIs.

## 3 RESULTS

Figures 3 and 4 summarize our results (Appendix A has the raw prediction results). Many of our features have overlapping information. POS tags include punctuation, BERT vectors have been shown to encode syntactic information (Hewitt & Manning, 2019) and ConTreGE vectors, built from constituency trees, encode some POS tags information. To detect brain regions sensitive to the distinct information given by a feature space, we build hierarchical feature groups in increasing order of syntactic information and test for significant differences in performance between two consecutive groups. We start with the simplest feature – punctuation, and then add more complex features in order: the effort-based metrics, POS and DEP tags, one of the ConTreGE vectors and the vectors derived from BERT (which can be thought of as a super-set of semantics and syntax). At each step, we test if the introduction of the new feature space leads to significantly larger than chance improvement in $R^2$.

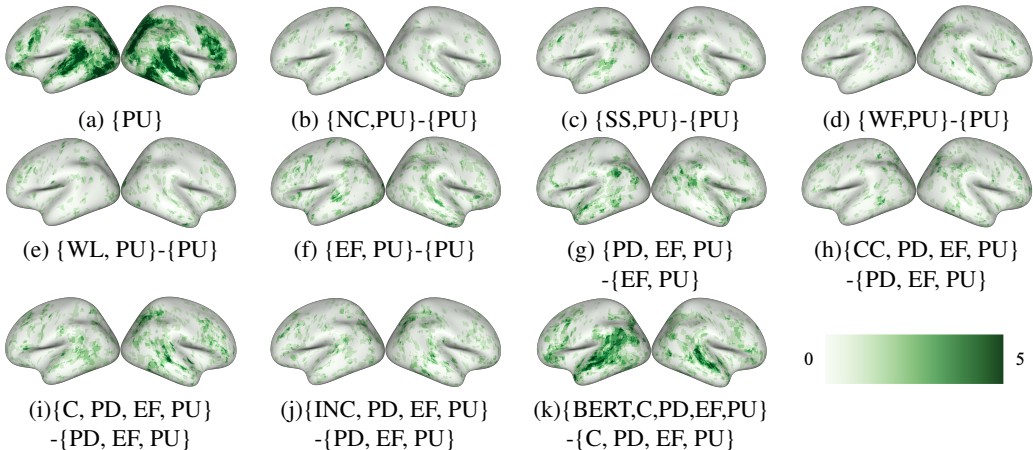

Figure 3: The first plot shows the number of subjects for which a given voxel is significantly predicted by punctuation ($p \leq 0.05$). The others show the number of subjects for which the difference in $R^2$ scores between two feature groups is significant ($p \leq 0.05$). Here, PU = Punctuation, NC = Node Count, SS = Syntactic Surprisal, WF = Word Frequency, WL = Word Length, EF = All effort-based metrics, PD = POS and DEP Tags, CC = ConTreGE Comp, C = ConTreGE, INC = InConTreGE, BERT = BERT embeddings and '{,}' indicates that these features were concatenated in order to make the predictions. '-' indicates a hypothesis test for the difference in $R^2$ scores between the two feature groups being larger than 0. The distinct information given by syntactic structure-based features is more predictive of brain activity than that given by effort-based metrics. The semantic vectors are also very predictive and many well-predicted regions overlap with those that are predicted by syntax.

**Syntactic structure embeddings are more predictive of brain activity than effort-based metrics**
Figures 3 (b)-(e) show that there are a small number of voxels that are predicted by the effort based metrics when taken in isolation. Figures 3 (f)-(i) indicate that although the information provided by the effort metrics combined is predictive of brain activity to some degree (when controlling for punctuation), there is still a considerable amount of structural information that is contained in the POS and DEP tags and in ConTreGE that predict additional portions of the activity. These results are made even clearer by Figure 4. Many voxels have significant increase in the $R^2$ scores (above what is predicted by the effort metrics) after including POS and DEP tags and ConTreGE. We also notice that ConTreGE Comp is not as predictive as ConTreGE, hinting that future syntactic information helps in predicting current brain activity. Additionally, InConTreGE is not as predictive as ConTreGE, suggesting that the top down parser might be generating partial parses that are not reflective of brain representations.

**ConTreGE results suggest that complex syntactic information is encoded in the brain** In this section we analyze the information in ConTreGE to interpret its brain prediction performance. We estimate how much of the constituency tree is captured by each feature by using it to predict the level

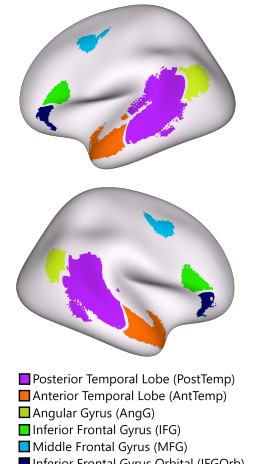

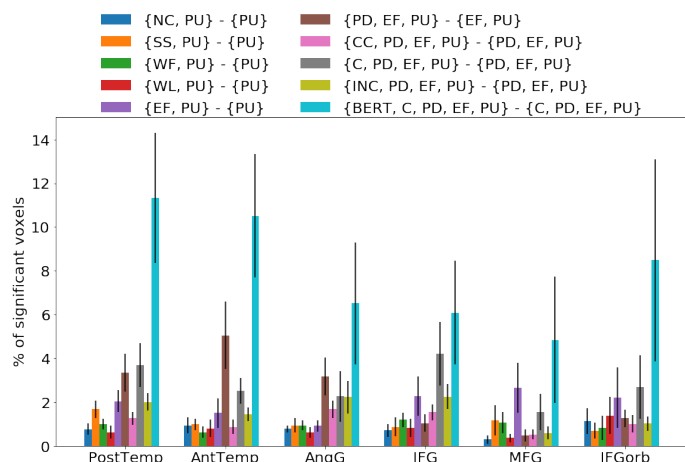

Figure 4: Region of Interest (ROI) analysis of the prediction performance. [Left] Language system ROIs by Fedorenko et al. (2010) from (3). [Right] Percentage of significantly predicted ROI voxels. Each bar represents the average percentage across subjects and the error bars show the standard error across subjects. We use the same abbreviations as in Figure 3 and see the same trends across ROIs.

| Feature | Level 2 | Level 3 | Level 4 | Level 5 | Level 6 | Level 7 | Level 8 | Level 9 |
|---|---|---|---|---|---|---|---|---|
| Most Popular Label % | 51 | 38.76 | 54.42 | 64.05 | 73.44 | 78.25 | 82.38 | 85.82 |
| Node Count | 51 | 42.6* | 55.45* | 64.01 | 73.4 | 78.23 | 82.4 | 85.82 |
| Syntactic Surprisal | 50.21 | 41.48* | 54.42 | 64.05 | 73.44 | 78.25 | 82.38 | 85.82 |
| Word Frequency | 51 | 41.56* | 57.13 | 66.58 | 76* | 80.68 | 84.78 | 88.16 |
| Word Length | 50.99 | 39.18 | 54.42 | 64.22 | 73.44 | 78.28 | 83.31 | 88.68* |
| POS and DEP tags | **92.23*** | **71.27*** | **67.06*** | **70*** | **77.51*** | 82.09* | 86.26* | 89.59* |
| ConTreGE Comp | 66.76* | 51.01* | 59.52* | 67.95* | 77.28* | 82.08* | 86.26* | 89.68* |
| ConTreGE | 52.91 | 45.42* | 57.37* | 67.29 | 76.87* | **82.3*** | **86.51*** | **90.39*** |
| InConTreGE | 52.46 | 45.59* | 57.61* | 66.75 | 76.13 | 81.08 | 85.27* | 89.1* |
| BERT Embeddings | 52.18 | 45.73* | 58.62* | 66.79 | 75.68 | 80.31 | 84.72 | 88.72 |

Table 1: 10-fold cross validation accuracies in predicting the ancestors of a given word. * denotes accuracies significantly above chance (tested using Wilcoxon signed-rank test, $p \leq 0.01$). POS and DEP tags best predict lower level ancestors while ConTreGE vectors best predict higher level ones.

N ancestor of a word (in its constituency tree). We vary N from 2 to 9 and train a logistic regression model for each N. Since POS tags are the level 1 ancestors of words, we start the analysis at N=2. Because there are many phrase labels, we group them into 7 larger buckets - noun phrases, verb phrases, adverb phrases, adjective phrases, prepositional phrases, clauses and other miscellaneous labels. Also, if a word's depth in its tree is less than N, the root is considered its level N ancestor.

Table 1 shows the results of this analysis. We use the constituency trees generated by the Kitaev & Klein (2018) parser. Given the skewed label distribution, the optimal strategy for a predictor that takes random noise as input is to always output the majority class ancestor at that level. Chance performance is thus equal to the frequency of the majority label. The effort-based metrics are not as predictive as ConTreGE at any level. POS and DEP tags are predictive of labels at all levels and produce the highest accuracies for lower levels. The InConTreGE vectors are not as predictive as ConTreGE or ConTreGE Comp, hinting that the top down parser might not be very accurate. ConTreGE is the best predictor of higher level ancestors but ConTreGE Comp is better than ConTreGE at predicting lower level ancestors. This may be because graph embeddings of a tree tend to capture more of the information near the tree's root (a random walk through a somewhat balanced tree is likely to contain more occurrences of nodes near the root). ConTreGE Comp vectors, created from shallow complete trees, likely over-represent lower level ancestors while ConTreGE vectors, created from relatively deeper trees, likely over-represent higher level ancestors. Given that ConTreGE is predictive of brain activity and contains information about the higher level ancestors of a word, this suggests that the brain represents complex hierarchical syntactic information such as phrase and clause structure.

**Syntax and semantics are processed in a distributed way in overlapping regions across the language system**    Our results indicate that syntactic and semantic information are processed in a distributed fashion across the language network. Most of the regions in the language system are

better predicted by the BERT embeddings after controlling for all our other feature spaces, and these regions overlap with the regions that are predicted by the syntactic feature spaces. While the BERT embeddings include both semantic and syntactic information, it is likely that the semantic information is at least partially predictive of the brain activity, given that we have already controlled for a lot of syntactic information.

## 4    DISCUSSION AND RELATED WORK

**Syntactic representations**    Apart from Brennan et al. (2012) and Hale et al. (2018), many others (Brennan et al., 2016; Henderson et al., 2016; Frank et al., 2015; Boston et al., 2008; Willems et al., 2015) use effort-based metrics to study syntactic processing during natural reading or listening. However, a few studies do explicitly encode syntactic structure: Wehbe et al. (2014) find that POS and DEP tags are the most predictive out of a set of word, sentence and discourse-level features. Moving away from popular approaches that are dependent on effort-based metrics, we extended the work of Wehbe et al. (2014) by developing a novel graph embeddings-based approach to explicitly capture the syntactic information provided by constituency trees. Our results showed that these explicit features have substantially more information that is predictive of brain activity than effort based metrics. Given these results, we believe that future work in this area should supplement effort-based metrics with features that explicitly encode syntactic structure.

**Syntax in the brain**    Traditionally, studies have associated a small number of brain regions, usually in the left hemisphere, with syntactic processing. These include parts of the inferior frontal gyrus (IFG), ATL and Posterior Temporal Lobe (PTL) (Grodzinsky & Friederici, 2006; Friederici, 2011; Friederici et al., 2003; Matchin & Hickok, 2020). However, some works point to syntactic processing being distributed across the language system. Blank et al. (2016) shows that significant differences in the activities of most of the language system are greater when reading hard to parse sentences than easier phrases.Wehbe et al. (2014) use POS and DEP tags to arrive at similar conclusions.

Previous work generally did not use naturalistic stimuli to study syntax. Instead, subjects are usually presented with sentences or even short phrases that have subtle syntactic variations or violations. Regions with activity well correlated with the presentation of such variations/violations are thought to process syntax (Friederici, 2011). Observations from such studies have limited scope since these variations often cannot be representative of the wide range of variations seen in natural language. This is possibly why such studies report specific regions: it might be that the reported region is particularly sensitive to the exact conditions used. By using one type of stimulus which evokes only one aspect of syntactic processing, syntax might appear more localized than it really is. Our results support the hypothesis that it is instead processed in a distributed fashion across the language system. We believe that our results have a wider applicability since we use naturalistic stimuli and we leave for future work the study of whether different syntactic computations are delegated to different regions.

Some studies have also doubted the importance of syntactic composition for the brain. Pylkkänen (2020) proposes that there is no conclusive evidence to indicate that the brain puts a lot of weight on syntactic composition, and that even though studies (some with effort-based metrics) have associated certain regions like the left ATL with syntactic processing, numerous studies have later shown that the left ATL might instead be involved in a more conceptually driven process. Gauthier & Levy (2019) showed that BERT embeddings which were fine-tuned on tasks that removed dependency tree-based syntactic information were more reflective of brain activity than those which contained this information. In contrast, our work uses purely syntactic embeddings to show that we can indeed significantly predict many regions of the language system. We attribute these differences in conclusions to our naturalistic stimuli and word-by-word evolving representations of syntax. Pylkkänen (2020)'s conclusions are mostly based on studies that present a phrase with just two words (like "red boat"). Gauthier & Levy (2019) use data averaged over entire sentences instead of modeling word-by-word comprehension. Since the syntactic structure of a sentence evolves with every word that is read, this approach is not necessarily adept at capturing such information.

Furthermore, our analysis of the syntactic information contained in various features highlighted that our ConTreGE vectors are good at encoding complex phrase or clause-level syntactic information whereas POS and DEP tags are good at encoding local word-level syntactic information. Several regions of the brain's language system were predicted by ConTreGE, hinting that the brain does indeed encode complex syntactic information. Another potentially interesting observation is that

including ConTreGE increases prediction performance in the PTL and IFG by more than when we include POS and DEP tags (Figure 4) but not for the ATL and the Angular Gyrus (AG). These observations very loosely support the theory by Matchin & Hickok (2020) - that parts of the PTL are involved in hierarchical lexical-syntactic structure building, the ATL is a knowledge store of entities and the AG is a store of thematic relations between entities. This is because ConTreGE encodes hierarchical syntactic information and word-level POS and DEP tags are very indicative of the presence of various entities (various types of nouns) and the thematic relations between entities (verbs associated with noun pairs). This hypothesis should be tested more formally in future work.

We also observe that ConTreGE is more predictive than ConTreGE Comp and InConTreGE with the latter two being very weakly predictive. Thus, future syntactic information appears to be very useful while predicting BOLD signals, indicating that that the brain anticipates the eventual sentence structure while reading to a more accurate extent than an incremental top down parser.

**Syntactic vs. semantic processing in the brain**    Finally, our results support the theory that syntax processing is distributed throughout the language network in regions that also process semantics. This theory is supported by other studies (Fedorenko et al., 2012; Blank et al., 2016; Fedorenko et al., 2020). However, Friederici et al. (2003) among others argue that syntax and semantics are processed in specific and distinct regions by localizing the effects of semantic and syntactic violations. Again, these differences might be due to the specialized stimuli and high statistical thresholds that only reject the null hypotheses in the regions with the strongest effect size, thereby precisely identifying small regions. A less conservative threshold might have revealed a more distributed pattern without leading to type I errors.

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
