# OpenReview forum: "Syntactic representations in the human brain: beyond effort-based metrics"
_ICLR.cc/2021/Conference — Reject_

### Official Review · AnonReviewer3 · 2020-10-21

**Rating:** 6
**Confidence:** 4

**Review:**

Summary

The manuscript focuses on understanding the features of syntax that are processed by different brain regions as captured by fMRI. The proposition is to move beyond effort-based metrics to subgraph embeddings for modeling syntactic structure. In addition, the approach focuses on natural reading and incremental models while a sentence is being read. I liked reading the paper and it reports interesting results. The advantage of the research is that it uses natural reading stimuli, but that also comes with disadvantages for supporting the conclusions of the manuscript (see details below).

General comments

The research questions and contributions are clearly written. I think the authors could milden some of the statements that are not considered to be really novel findings of the present manuscript.

It is known that syntactic processing and semantic processing are mixed and that several brain regions contribute to constitute an understanding of language (see e.g. 1,2). It is also known that the brain not only processes structure seen thus far but also predicts future structure. However, this, and other measures, such as syntactic violations, information gain, etc. has been found already in the early studies (see 3,4,5) and more recently in natural reading (see 6). Maybe the authors could be a bit more specific in what exactly is novel in the present manuscript.

Methods

I did not fully understand the data split for training and testing. It states:  “is first split into 4 contiguous and equal
sized folds. Each model uses three folds of the data for training and one fold for evaluation.” Now, if the data is from natural reading of text, does this mean that you are using samples that occur both before and after a particular word is being read? If so, does this affect the results as evidence from other (latter) parts of the sentences can be used to draw predictions for words earlier in the sentence? Prior to this, it is stated that you only use x-1,…x-4 for prediction and that “(to form a prediction for the entire experiment in which each time point
is predicted from a model trained without the data for that time point)”. But is there now a chance that something in x+n would be in the training data for the models? Wouldn’t this mean that some data would “leak” in x+n and would be in the training set for a particular sample? Maybe I didn’t correctly understand the setting, but I would like to see a better explanation of how exactly the split is done and test and training sets are constructed, and why this is the correct way for the particular analysis conducted.


There are many models tested and I hope the authors are correcting for multiple testing; it is stated that Benjamni-Hochberg False Discovery Rate correction is applied, but it is not entirely clear how this is done.

I did not understand why principal component analysis (PCA) was used to reduce the dimensionality to 15. Why would the dimensionality need to be the same? How does this affect the results?

Conclusions

The conclusions of the paper rely on the advantages of naturalistic sentence reading, but the claims that more controlled experiments that have revealed specific regions responsible for syntactic processing would be less important, are not supported by the results presented in the manuscript. There was nothing in the study setting that would have controlled for other effects. For example, what if there were some other factors in the sentences that explain the effects. What if the other model predicts some other features that are correlated with syntax structure, such as more complex words, words that carry more information, more frequent, rare, short, or long words? I think these cannot be fully excluded and I would have liked to see some other results than only the predictions of the models to confirm that the prediction study can be considered valid. Without a proper control condition or other analysis of the data, it is not evident that all of the conclusions hold.

[1] https://www.sciencedirect.com/science/article/pii/S1053811915011064?casa_token=uVeVeMM5HCwAAAAA:Uj7IsuC-Kf6xwHQ-Rgs8RhxTl8A_PID_2fVStTqnuTA8Lshjb8Lil-iDOhyHJgADMGhHtjDM8kE

[2] https://www.nature.com/articles/s41467-019-08848-0

[3] https://www.sciencedirect.com/science/article/pii/S1053811916001592?casa_token=neHunii1ASwAAAAA:P1lkCeo5kDULBuy_6VXAjFN2gx2xK-algZXjXWaRgNxpnhGnWA3y_Vh_ey4TvynL9-CZOHHrkI0

[4] https://www.sciencedirect.com/science/article/pii/S0093934X14001515

[5] https://www.mitpressjournals.org/doi/abs/10.1162/089892903322370807?casa_token=SqKOci8rCScAAAAA:rvso0ntTEMzNym_FhyH-ylSilW0qRHjEiBLrFbXlV703WHLOTTZ4G3xHVHhrxRiIBXLB9PqadvV3

[6] https://www.nature.com/articles/s41598-020-63828-5

[7] https://www.sciencedirect.com/science/article/pii/S1053811916001592?casa_token=WtO6Twcn9yEAAAAA:6aeIl_agLqFBsaViu4Audy7KdHJKtELPp6TF2KT0stCpng80APTfD7V1rZu0gpzqDqFp8ZKHy2E

---

> ### Author Response · Authors · 2020-11-16
> **Reply to Reviewer 3**
>
> Thank you for your detailed review and for the suggestion to milden the statements. We followed this suggestion in the updated version of the manuscript by softening our contribution and expanding on some of them to give more details and we hope that these changes help address the other concerns of the reviewer.
>
> We think that the statement of “It is known that syntactic processing and semantic processing are mixed and that several brain regions contribute to constitute an understanding of language (see e.g. 1,2)” is too strong. In fact, the Blank et al. (2016) paper which we cite in our manuscript represents a new way of thinking about the language system that is not in agreement with many existing theories that posit that syntax and semantics are processed in different locations (see Fedorenko et al. (2020) for a recent discussion). We believe that one of the contributions of our paper is to show that this result generalizes (and therefore add confidence in it) by showing it holds in a new experiment and through an entirely different paradigm (building an encoding model with syntactic features instead of looking at activity across conditions). In science, we can only start trusting a finding once we see it being replicated and generalized to new conditions and tasks, making replications and generalizations valuable. What is also novel in our manuscript is the use of explicit syntactic feature spaces which go beyond univariate effort-based metrics such as node count, surprisal, information gain etc, and explicitly represent the syntactic information. This has not yet been done in the field of syntax processing during language processing (even though people have been using *semantic* feature spaces to encode the meaning of language for many years).
>
> Thank you for the question about our multiple comparison correction. We were correcting at the brain level (each participant has around 30000 voxels). We agree with the reviewer that we need to also account for the multiple feature spaces used, and we repeated the analysis by grouping all the voxel level results (for all feature spaces and for all subjects) and then doing FDR correction once. This results in one threshold for all the results, which leads to small variations in the results, but the general pattern remains the same. We have updated the plots in the manuscript.
>
> We did not intend to say something that suggests that controlled experiments were less useful, in fact, we think that both controlled and naturalistic experiments are complimentary in the scientific endeavor. We computationally controlled for the effects of word frequency and word length by explicitly including these measures in the early groups of feature spaces. Then, if the additional feature spaces such as ConTreGE still predict brain activity even after controlling for effects such as word frequency, the assumption is that these additional feature spaces must contain other types of information that are predictive. We agree with the reviewer that there is perhaps some correlation that cannot be removed, and we have included this limitation in the updated manuscript.

---

> ### Author Response · Authors · 2020-11-16
> **Reply to Reviewer 3 continued**
>
> Here is a detailed explanation of the training and testing setup. Our split and methods are consistent with other work in building encoding models for naturalistic continuous stimuli (see Wehbe et al. (2014)).
> - Due to the resolution of fMRI, brain images are acquired at a rate of 2s per image, while each word is read for 0.5 seconds.
> -The text was presented to the subjects in 1291 TRs.
> - In each TR (1 TR = 2s), 4 tokens were presented one by one (each token was presented for 0.5s).
> -All of our features are word-level features. Thus, each token has a feature vector associated with it. Let the dimensionality of the feature vector be $d$.
> -The brain activations are measured at the level of TRs. Thus we have 1291 activation values (denoted by $Y = [y_1, y_2, …, y_{1291}]$ in the paper).
> -Now, since 4 words were presented in a TR, we have 4 feature vectors associated with each TR. We reduce these 4 feature vectors into one vector by just summing them. This becomes the $d$ dimensional aggregated feature vector $x_i$ that is associated with $y_i$. fMRI is recording slowly varying activity that is much slower than the pace the words are presented at. It is thus not possible to distinguish the activity related to individual words, and our analyses necessarily have to be at the aggregate word level (again this is consistent with other naturalistic imaging work).
> -Thus, we now have a set of input vectors $X = [x_1, x_2, …, x_{1291}]$ and a set of output vectors $Y = [y_1, y_2, …, y_{1291}]$ and we want to analyse if we can accurately predict $y_i$ using $x_i$. However, because of the lag in the fMRI response, we instead try to predict each $y_i$ using $x_{i-1}, x_{i-2}, x_{i-3}$ and $x_{i-4}$. This is a common approach as mentioned in the manuscript. Thus, the final set of input vectors used to fit the models is $X_{lag} = [xWithLags_1, xWithLags_2, … , xWithLags_{1291}]$ where $xWithLags_i = [x_{i-1}, x_{i-2}, x_{i-3}, x_{i-4}]$. The first few $xWithLags_i$ do not have defined $x_{i-1}, x_{i-2}, x_{i-3}, x_{i-4}$ and we just use zero-padding in these cases.
> - We use the $R^2$ metric to measure the correctness of these predictions. This metric is computed as follows:
>     - The $X_{lag}$ and $Y$ sets are broken up into 4 contiguous and equal-sized folds (only the last fold has 1 less TR than the other folds). Let’s call them $X1, X2, X3, X4$ and $Y1, Y2, Y3, Y4$. Since they are contiguous, $X1 = [xWithLags_1, xWithLags_2, …, xWithLags_{323}]$, $X2 = [xWithLags_{324}, xWithLags_{325}, ... , xWithLags_{646}]$, $X3 = [xWithLags_{647}, xWithLags_{648}, ... , xWithLags_{969}]$, $X4 = [xWithLags_{970}, xWithLags_{971}, …, xWithLags_{1291}]$ and similarly for $Y1, Y2, Y3, Y4$.
>     - Then four separate models are built, each trained using 3 folds of data and evaluated on the remaining fourth fold. For example, the first model is trained using $X2, X3, X4$ concatenated and $Y2, Y3, Y4$ concatenated. It is then asked to make predictions with $X1$ being the input. Let us call these predictions $Z1$. The second model is then trained using $X1, X3, X4$ and $Y1, Y3, Y4$ and asked to make predictions with $X2$ being the input to get $Z2$. Similarly, we get $Z3$ and $Z4$.
>     - Finally, $Z1, Z2, Z3$ and $Z4$ are concatenated to obtain $Z$. The set of predictions $Z$ has length 1291 i.e. it contains predictions for the entire experiment. This final set of predictions $Z$ is compared to $Y$ to obtain the $R^2$ score.
>     - This setup does not lead to any intentional data leaks. We were not sure about what you meant by there being a leak. However, while pondering over this question, we did realize that there might be some sort of unintentional leak because of the lag in the HRF, leading to some close $y_i$ being very similar to each other. Thus, we obtained a new set of results by removing data from the 5 TRs which precede and follow the test fold from the training set of folds for each model. For example, this means that the second model is trained using $X1’, X3’, X4$ concatenated where $X1’ = [xWithLags_1, xWithLags_2, … , xWithLags_{318}]$ and $X3’ = [xWithLags_{652}, xWithLags_{653}, …, xWithLags_{969}]$ and similarly $Y1’, Y3’$ and $Y4$. This removal of data leads there being no leaks even due to the lag in the HRF. Our results do not change much even after performing this correction as is evident in the revised manuscript.
>
> The original dimensionality of BERT embeddings is 1024, which leads to a 4096-dimensional input vector as explained above after considering the delays and we have less than 1000 time points in the training set. We therefore reduced the dimensionality to avoid overfitting and chose 15 because it is the dimensionality of ConTreGE.

---

### Official Review · AnonReviewer4 · 2020-10-28
**Looking for syntax correlates in fMRI data**

**Rating:** 8
**Confidence:** 4

**Review:**

This paper presents a neuroimaging study investigating the way syntax is represented. The authors compare models that encode syntax with fMRI data. They find that syntax and semantics are computed/represented in overlapping brain regions and that "complex" syntactic information is decodable.

I liked this paper and think it is valuable in, amongst other things, furthering the theoretical position that the dichotomy between semantics and syntax is a more conceptual/high-level one than can be found at the level of neuroimaging data.

The authors give an exposition of their research questions, however I think these can be phrased even more clearly in some cases. For example, for Q1 do they mean for humans in general (as in in the brain) or do they mean us as scientists researching human cognition? For Q2, do they mean they will use a model-based fMRI analysis? For Q3, is this multivariate or univariate? Just adding those short phrases or words to the research questions will help situate the reader, in my opinion.

Why is it in and of itself surprising that (complex) syntax is encoded in the brain? In other words: "Several regions of the brain’s language system were predicted by ConTreGE, hinting that the brain does indeed encode complex syntactic information." — why would "the brain" not? Please do not get me wrong, this is obviously important/required to be shown but the research herein actually has even more value (or could have) and can be framed and discussed as such. Surely, the interesting results (since we know from other sources and common sense that syntax is, has to be, encoded in the brain somewhere since it plays a role in cognition) is the actual relationships of the results to the overarching theory and should be foregrounded more.

A potentially useful theory paper is, and which might interest the authors: https://doi.org/10.1162/jocn_a_01552

Minor point: the in-text citations would look better without double brackets — which is easy to fix in LaTeX.

---

> ### Author Response · Authors · 2020-11-16
> **Reply to Reviewer 4**
>
> Thank you for your positive review. We have updated the paper to clarify that we mean how can we as scientists researching human cognition construct these embeddings in Q1, that we are contrasting these different embeddings when using them as input for encoding models for fMRI in Q2 and that we contrast our embeddings of structural syntactic information with embeddings of semantic information in Q3.
>
> We understand why the reviewer is questioning the surprising nature of the result that complex syntactic structure is encoded in the brain. While this sounds intuitive, many recent studies have doubted the role of syntactic processing during comprehension. We have cited some of these works in the discussion including Pylkkanen (2020) and Gauthier and Levy (2019).  Pylkkanen (2020) for example argues that there is no conclusive evidence to indicate that the brain puts a lot of weight on syntactic composition. Thank you for the suggested paper.
>
> We were not sure what you meant by the double brackets, we are using the provided ICLR template. We would be happy to fix the bracketing if you could point us to an example of where this needs to be done!

---

> > ### Comment · AnonReviewer4 · 2020-11-16
> > **LaTeX**
> >
> > Thanks for the reply. I have nothing to add.
> >
> > Here is an example for the mis-used command you asked for:
> > > (ex. Brennan et al. (2016); Henderson et al. (2016); Frank et al. (2015); Boston et al. (2008); Willems
> > et al. (2015))
> >
> > Ideally you want something more like:
> >
> > > (ex. Brennan et al. 2016; Henderson et al. 2016; Frank et al. 2015; Boston et al. 2008; Willems
> > et al. 2015)
> >
> > Easily doable if you look up what natbib (if you are using that — mutatis mutandis for raw bibtex, etc.) is needed, e.g.: https://gking.harvard.edu/files/natnotes2.pdf

---

> > > ### Author Response · Authors · 2020-11-17
> > > **Corrected**
> > >
> > > Thank you for pointing that out! We have fixed the formatting in the manuscript.

---

### Official Review · AnonReviewer1 · 2020-10-28
**Official Blind Review**

**Rating:** 4
**Confidence:** 3

**Review:**

This paper aims to propose a parse tree embedding that correlates with brain activations better than existing measures on sentences. It is an extremely important topic as it can draw the link between Artificial NN and real NN on the problem of syntactic processing.
However, the paper leaves with a major quesiton: why?
Why the proposed parse tree embedding model is a good model. There is a wide range of models embedding parse trees, e.g. RecursiveNN, TreeLSTM and Distributed Tree Kernel. All these models are embedding trees in different ways. Why the proposed model should be closer to the brain activity? This should be definetly clarified in the description of the parse tree embedder.

---

> ### Author Response · Authors · 2020-11-16
> **Reply to Reviewer 1**
>
> Thank you for the review. From our understanding of this review, it seems like the need for a new way to encode constituency trees is being questioned. To answer this question, it should be noted that our method of encoding subtrees allows the embeddings to be purely syntactic and it has the ability to encode the structure of any subtree (even incomplete ones). RecursiveNNs and TreeLSTMs are architectures that can be used to generate constituency trees and/or to compute embeddings of the constituents of a sentence. However, these embeddings contain semantic information. Thus, such architectures cannot be used for our use case since we are looking to compute purely syntactic representations that encode the structures of the subtrees. Distributed Tree Kernels (DTKs) can certainly be used to obtain such representations of subtrees of constituency trees. However, using them did not yield good results when compared to ConTreGE.
>
> The result is here (anonymous link) - https://drive.google.com/file/d/1s_7Upt_B6svVPYSZJtAA-0pxTr7tCkgn/view?usp=sharing
>
> To get this result, we embedded the incomplete subtrees (used to construct ConTreGE vectors) by employing the embedding methods used in DTKs. 8192 dimensional embeddings were computed (same as in the original DTK paper) using fast shuffled convolutions and the lambda parameter was set to 0.4 (this value of lambda was used since the original DTK paper showed that it produces some of the best results). Then, PCA was used to reduce the dimensionality of these embeddings to 178 dimensions (retains 80% of the variance). Finally, we tested if the inclusion of this feature led to a significant improvement in the $R^2$ values after controlling for punctuation, the effort-based metrics and POS and DEP tags (similar comparison as in figure 3 (i) but we use the new DTK-based vectors instead of ConTreGE here).
>
> Also, we compare our new syntactic embeddings with other established syntactic features rather than other embedding techniques because we believe that this question is more important. We do not argue that our embedding technique is perfect but we do believe that it can be an important asset to studying syntactic processing and the paradigm itself is extensible and novel, lending itself to usage in future studies.
>
> Now, the second question being asked is why the proposed model should be closer to the brain activity. We have tried to answer this question by showing that the ConTreGE vectors encode higher level syntactic information that is not encoded by the other syntactic features. Since this information might be processed by the brain too, this might explain why these embeddings are better at predicting brain activity.
>
> It must be noted that, to the best of our knowledge, no other work has explored building comprehensive, purely syntactic feature spaces to study the brain. One of our motives is to start this line of research.

---

### Official Review · AnonReviewer2 · 2020-10-28
**Interesting work, but conclusions yield minor impact / may not follow from results**

**Rating:** 5
**Confidence:** 4

**Review:**

This paper derives various types of graph embeddings to encode aspects of syntactic information that the brain may be processing during real-time sentence comprehension. These embeddings, along with indicators of punctuation, POS and dependency tags, and BERT embeddings, are used to predict brain activity recorded via fMRI. The authors argue that this is an improvement over use of effort-based metrics to predict brain activity, as these embeddings contain richer information than is captured by distilling down to a single measure of effort. They show that various brain regions are significantly better predicted by the syntactic embeddings than by the effort-based metrics and POS+dependency indicators. BERT embeddings, however, prove to be a better predictor (than syntactic and other predictors) across much more substantial areas of activity.

I'm all for leveraging representations from NLP models to ask questions about the brain, and some of the patterns identified here are interesting, but I don't think this paper has arrived at sufficiently impactful takeaways to merit publication as yet. The main concrete conclusion drawn from these analyses is that complex syntactic information is encoded in the brain. But this really isn't a particularly disputed claim, so it's definitely underwhelming as a takeaway. The sensitivity of the brain to hierarchical syntax is also an underlying assumption of syntactically-grounded effort-based metrics, so although these graph embeddings capture richer information, if they only give us the conclusion that the brain is doing syntax, then they have not really given us new information. A related conclusion made in the paper is that syntactic information is represented in a distributed fashion, but the effort-based metrics seem to suggest a similar conclusion (if I'm correctly interpreting Fig 3f), so again this is not unique to the proposed representations. I do recognize that the proposed syntactic representations are predictive of activity over and above the effort-based metrics in some voxels, but it's not clear exactly what we learn from this fact.

A secondary conclusion made in the paper is that regions that process syntax are not specialized for syntax. This conclusion seems to be made based on the fact that BERT embeddings are stronger predictors than the syntactic embeddings in many of the regions in which the syntactic embeddings outperformed other predictors. However, as the authors acknowledge, BERT embeddings also encode syntactic information, and this makes it more difficult to interpret this pattern of results. It seems that the stronger performance of BERT embeddings could just as easily be attributable to better/richer encoding of relevant syntactic information as to encoding of semantic information.  What the results seem to show is simply that the BERT embeddings are better predictors of brain activity than any of the other representations used, so the question then raised is what exactly the BERT embeddings capture that the brain activity is also sensitive to.

I'll also say that a downside of the graph embeddings is that they seemingly reduce transparency relative to the effort-based metrics -- I'm certainly willing to believe that they encode richer information, but it doesn't seem clear precisely what information they are adding.

All in all, I think this is an interesting line of work, but I'm not convinced that we come away having learned something impactful, both because some of the proposed conclusions answer questions that aren't really at-issue / aren't really uniquely addressed by the proposed representations, and also because some conclusions are drawn that don't seem to follow clearly from the observed results.

---

> ### Author Response · Authors · 2020-11-16
> **Reply to Reviewer 2**
>
> We thank the reviewer for their detailed review. Regarding the contributions of the paper, we would like to point out one contribution that appears to be understated - the new methodological approach to study syntax in the brain. Even though many researchers have investigated the brain basis of syntax, no one yet to our knowledge has explicitly used syntactic representations. We consider this work to be a way to introduce other neuroscientists to this approach in the hope that they will partake in creating new syntactic feature spaces to test their syntactic hypotheses. Studying syntax in fMRI and with natural language is a complicated experimental endeavor with many limitations and challenges which might not be obvious to the reader. Through this work, we are able to propose a new way to study syntax, and leave the door open for other neuroscientists to propose other feature spaces encoding structural information that is constructed based on other hypotheses about syntax.
>
> While it is true that effort-based metrics are easy to interpret, we have shown that they do not predict much brain activity after controlling for sentence boundaries, and that explicit feature spaces that encode syntactic information are more predictive of brain activity. Thus, it does not seem justified to discard more predictive yet possibly less interpretable representations in favour of these existing metrics. This would be akin to not using deep neural networks just because their workings are not entirely interpretable. Even given that concession, our feature spaces are not all non-interpretable: the part of speech and dependency role features are by definition interpretable, and the parse-tree embeddings are effectively a transformation of the structure of the constituency tree of a sentence, which is a very well established theoretical linguistic construct. Moreover, we try our best to deconstruct the graph embeddings we propose using the ancestor prediction analysis. Encoding objects and structures such as trees is a difficult and open problem in machine learning and AI. As methods progress, we might one day have more expressive or more easy to interpret graph embeddings of parse trees. Our framework is extensible and allows for future analyses to use different embeddings.
>
> We thank the reviewer for the discussion of the novelty in this paper. We believe the following are two valuable contributions of this paper:  (a) it proposes a new methodology for computational cognitive neuroscience with a new way to look at an important question and (b) it confirms some of the findings in the field (which are more controversial than stated by the reviewer) in a new experiment and using another paradigm. We believe both of those contributions can contribute to the progress and reproducibility of the cognitive neuroscience of language.

---

> > ### Comment · AnonReviewer2 · 2020-11-24
> > **Reply to authors**
> >
> > Thank you to the authors for your detailed replies. I want my score to acknowledge that it is certainly noteworthy that the proposed representations, as well as the BERT representations, are more predictive of brain activity than are effort-based metrics. This is surely meaningful, and surely points us toward some interesting insights about what information is reflected in the relevant brain activity. I am going to increase my score accordingly.
> >
> > However, I remain hesitant because although the better predictive power surely means something, I still don't feel the paper in its present form gives us a clear/confident answer about *what* the findings mean about the brain's representation of syntax -- again, in large part because of the fair amount of opacity in both the graph embeddings and the BERT embeddings. Since the aim of this line of work is precisely to shed light on how the brain works, this seems an important shortcoming, and one that I would love to see addressed more substantively to improve the paper's ultimate impact. Along the same line, the authors' argument about taking a representation-based approach to studying syntax in the brain is an interesting one, but I think that it will be easier to sell the value of this approach if you can give a clear illustration of the precise insights that this approach affords beyond existing methods. For this reason I am setting my score at 5.

---

> > > ### Author Response · Authors · 2020-11-25
> > > **Reply**
> > >
> > > Thank you very much for increasing the score! We would like to point out that we do test three different types of graph embeddings with each trying to answer a different question. The ConTreGE Comp vectors allow us to investigate if the brain is concentrating on local syntactic information; the ConTreGE vectors indicate whether the brain anticipates future syntactic structure and whether it processes phrase-level syntactic structure; and finally, the InConTreGE vectors can tell us whether the brain could be computing several possible top down partial parses that can derive the words seen thus far. These specific questions can indeed be investigated using our vectors. The fact that most of our features are predictive across the language network might make a reader question the importance of our embeddings since specific regions of the brain cannot be associated with different syntactic processes. However, if it is true that the entire network processes syntax as recent research suggests (see Fedorenko et al. (2020)), these findings might indeed be representative of how the brain truly processes syntax.
> > >
> > > Moreover, our method of encoding constituency trees can be used to study other syntactic hypotheses as well. For example, one could use them to encode trees generated using different parsers to determine which parser produces trees that are closest to the brain’s representations. They could also be used to examine the “range” of the syntactic information which is represented in the brain. For example, we have tried varying the height of the subtrees (by truncating them) to see if a certain height leads to better predictions (these results were not included for the sake of brevity). This could inform us about how the brain processes very complicated sentence structures and if it only represents more recent syntactic information in such cases. These types of hypotheses that rely on varying parse trees cannot be effectively studied using the current set of effort-based metrics but they can be studied by building graph embeddings using our techniques.
> > >
> > > Finally, it must be noted that computational techniques can be used to mainly generate hypotheses or give some amount of initial validation. Only controlled experiments with carefully designed stimuli could possibly be used to strongly support/confirm hypotheses about the brain although some such studies have also shown variable results. In this context, we believe that our paper can definitely aid in hypothesis generation and provide initial validation of a hypothesis about syntactic processing. This can help researchers weed out hypotheses that are very likely to be false and concentrate on confirming those that are likely to be true.

---

### Author Response · Authors · 2020-11-16
**General reply to reviews**

We thank all the reviewers for their comments and suggestions for the manuscript. We have run a few additional analyses to address some of the concerns of the reviewers and we show that they have not changed the conclusions of the paper.

One major theme across the reviews concerns the contributions of this paper. This paper calls for the use of syntactic structure embeddings to study syntax in the brain, and thus draws new connections between AI and neuroscience in the area of syntactic processing, in a way that did not exist before. Indeed, neuroscientists have not to our knowledge studied the representation of syntax. Instead, they typically look for areas that are correlated to the effort related to processing syntax. This paper therefore is a new methodological contribution that calls for a shift in perspective in the cognitive neuroscience of syntax. It shows that AI can directly contribute to the neuroscience of language, in the form of encoding hypotheses about syntactic processing in embeddings that explicitly capture syntactic information (and not just univariate effort based metrics).

Multiple reviewers were concerned that the results about the distributed representation of syntax in the same regions where semantics is processed were not “new”. There are multiple competing theories about how syntax is represented in the brain in the literature, with one group considering that syntax and semantics are represented separately, and the other considering that they are represented together. Both groups have empirical evidence supporting their conclusions. How can the scientific community decide between competing theories? By designing new experiments that test these theories in different settings and see what conclusions these additional studies point to (when considered together as a body of work and not at the individual study level). Our current paper fits in this philosophical view of science and offers a generalization of the theories of the second group (semantics and syntax are processed jointly) in a new experiment, that is naturalistic (and therefore closer to how the brain processes information in real life than controlled conditions) and using an entirely different method that relies on a computational model (explicitly encoding the syntactic information in syntactic embeddings).

---

### Decision · Program_Chairs · 2021-01-07
**Final Decision**

**Decision:**

Reject

**Comment:**

This paper explores the brain's activity in response to language, specifically targeting the signatures of syntax in the brain.  The authors specifically investigate the signatures of specific syntactic elements against the "typical" effort based syntax measures from some previous work.

The title and abstract of the paper are clear and compelling, but the text of the paper muddies the message and this was expressed in the reviews.  There may be some debate in the literature as to if syntax and semantics are dissociable, and to what degree we can actually measure syntax in the brain, but I (and your reviewers) have trouble believing that any one actually thinks there is *no* syntax representations in the brain.  Certainly this is not a claim made by either the Federenko or Pylkkanen papers the authors cite. Federenko says "lexico-semantic and syntactic processing are deeply inter-connected and perhaps not separable" but doesn't claim that the brain doesn't "do" syntax. Pylkkanen says "Syntax in the brain is necessary to explain the fact that humans are exquisitely skilled at judging syntactic well-formedness, even for sentences that have no coherent meaning."

I suggest this paper either rephrase the arguments, more clearly articulate the issues they wish to address, or find another venue where the reviewers might be more read to debate *if* syntax is encoded in the brain.  That seems outside of the scope of ICLR.

---

> ### Author Response · Authors · 2021-02-16
> **Response to meta-review**
>
> Thank you for your meta-review. We would just like to clarify that we only said that there are some papers which question the level of importance of syntactic composition for the brain, including Pylkkänen (2020). Furthermore, this was a rather secondary point in our paper and responses. The main argument is about the importance of explicitly representing syntax and the fact that semantics and syntax are represented in the same regions. Also, we use the papers by Fedorenko and colleagues to in fact support the hypothesis that syntactic representations can be seen in brain data and that the regions which process them are not very distinguishable from those that process semantics. We are not sure how this came across as meaning that these papers support the theory that there are no syntax representations in the brain.